

# Exceptionally preserved 'skin' in an Early Cretaceous fish from Colombia

Andrés Alfonso-Rojas[1,*] and Edwin-Alberto Cadena[1,2,*]

[1] Facultad de Ciencias Naturales, Grupo de Investigación Paleontología Neotropical Tradicional y Molecular (PaleoNeo), Universidad del Rosario, Bogotá, Colombia
[2] Smithsonian Tropical Research Institute, Panama City, Panama
[*] These authors contributed equally to this work.

## ABSTRACT

Studies of soft tissue, cells and original biomolecular constituents preserved in fossil vertebrates have increased greatly in recent years. Here we report preservation of 'skin' with chemical and molecular characterization from a three-dimensionally preserved caudal portion of an aspidorhynchid Cretaceous fish from the equatorial Barremian of Colombia, increasing the number of localities for which exceptional preservation is known. We applied several analytical techniques including SEM-EDS, FTIR and ToF-SIMS to characterize the micromorphology and molecular and elemental composition of this fossil. Here, we show that the fossilized 'skin' exhibits similarities with those from extant fish, including the wrinkles after suffering compression stress and flexibility, as well as architectural and tissue aspects of the two main layers (epidermis and dermis). This similarity extends also to the molecular level, with the demonstrated preservation of potential residues of original proteins not consistent with a bacterial source. Our results show a potential preservation mechanism where scales may have acted as an external barrier and together with an internal phosphate layer resulting from the degradation of the dermis itself creating an encapsulated environment for the integument.

## INTRODUCTION

Exceptional preservation in the fossil record is expressed in a wide range of structures including hair, cells, blood vessels, claw sheaths, feathers, pycnofibers, muscle remains, skin and even the potential remains of original biomolecular constituents (DNA, proteins, lipids) (*Lingham-Soliar & Plodowski, 2010*; *Cadena, 2016*; *Cadena & Schweitzer, 2012 Cleland et al., 2015*; *McNamara et al., 2018a*; *Schweitzer, 2011*; *Wiemann et al., 2018*; *Bailleul et al., 2020*) associated with these structures. The skin is the largest organ of the a vertebrate body, which encloses or covers their entire body. Numerous integumentary derivatives are located within the epithelial sheet itself (glands) or extend above its surface (teeth, scales, feathers, hairs, etc.) (*Chernova, 2009*). The skin of vertebrates and its derivate structures has been shown to have high preservation potential in the fossil record, and has been reported in dinosaurs, pterosaurs, snakes, frogs and birds (*McNamara et al.,*

Corresponding author
Edwin-Alberto Cadena,
edwin.cadena@urosario.edu.co

*2018a*; *McNamara et al., 2016*; *McNamara et al., 2009*; *McNamara et al., 2018b*; *Varejão et al., 2019*). Similarly, fishes are also covered by a relatively flexible skin, which in almost all extant and extinct groups is associated with hard scales composed of collagen I, calcium salts (*Sionkowska & Kozlowska, 2014*), ganoine and cosmine. Preservation of skin in fossil fish has been documented in many Konservat Lagerstätte sites, including the Messel Formation, Germany (*Micklich, 2002*), Huajiying and Yixian formations (*Xu et al., 2020*); and Romualdo Formation (previously Santana Formation) of northeastern Brazil (*Kellner et al., 2013*; *Maisey, 1991*; *Martill, 1989*; Fig. 1D).

Despite the abundant recent discoveries of fossil vertebrates from the Cretaceous of Colombia (*Cadena, 2015*; *Cadena & Parham, 2015*; *Cadena et al., 2019*; *Carballido et al., 2015*; *Maxwell et al., 2019*; *Noé & Gómez-Pérez, 2020*; *Páramo-Fonseca et al., 2016*; *Vernygora et al., 2018*), the exceptional preservation of soft tissue or their potential original components is still rarely reported for most of them, with the exception of the recently described gravid marine turtle from the Early Cretaceous of Villa de Leyva (*Cadena et al., 2019*). Here we report a caudal fragment of an aspidorhynchid fossil fish recovered from the lower segment of the Paja Formation from Zapatoca, Santander, Colombia (Figs. 1A–1C) that constitutes the first specimen of the paleontological collection at Universidad del Rosario in Bogotá. We have applied multiple analytical techniques to interrogate the degree of preservation of its skin, including some of their potentially original biomolecular constituents. Our finding not only expands the worldwide record of skin preserved in Cretaceous vertebrates, but also constitutes the most equatorial example of it (Fig. 1D) considering that Colombia has barely changed its latitude since the Early Cretaceous (Fig. S1).

## MATERIALS & METHODS

**Fossil material Collection and Geological framework**. UR-CP-0001 specimen was collected by E-A. Cadena in 2016, during a short expedition to Zapatoca. The fossil was found approximately 100 m north-west from the Radio Lenguerke station antenna region, Zapalonga locality (6°48′28.94″N, 73°16′08.23″W, 1703 m) (Fig. 1A), inside a gray-purple sequence dominated by mudstones with abundant occurrence of large concretions and interbedded layers of fossiliferous limestones (Fig. 1B). This sequence represents the most basal member of the Paja Formation in this zone, a few meters above the last limestone bank of the underlying Rosablanca Formation. Approximately 35 m of stratigraphic column were measured and described (Fig. 1C).

The fossil was collected using sterile nitrile gloves and wrapped in aluminum foil, and placed in a plastic bag with silica gel small packets to control humidity. To avoid any contamination, the fossil has not been treated mechanically or chemically and always has been manipulated using sterile nitrile gloves for measurements, photography or sampling for analytical studies. Fieldwork and laboratory experiments permit granted by the Comité de ética and the Dirección de Investigaciones of the Universidad del Rosario (IV-FCS018). **Specimen photography, internal observation and measurements**. General views of UR-CP-0001 specimen were obtained using a Leica-EZ4-HD and Nikon SMZ1270

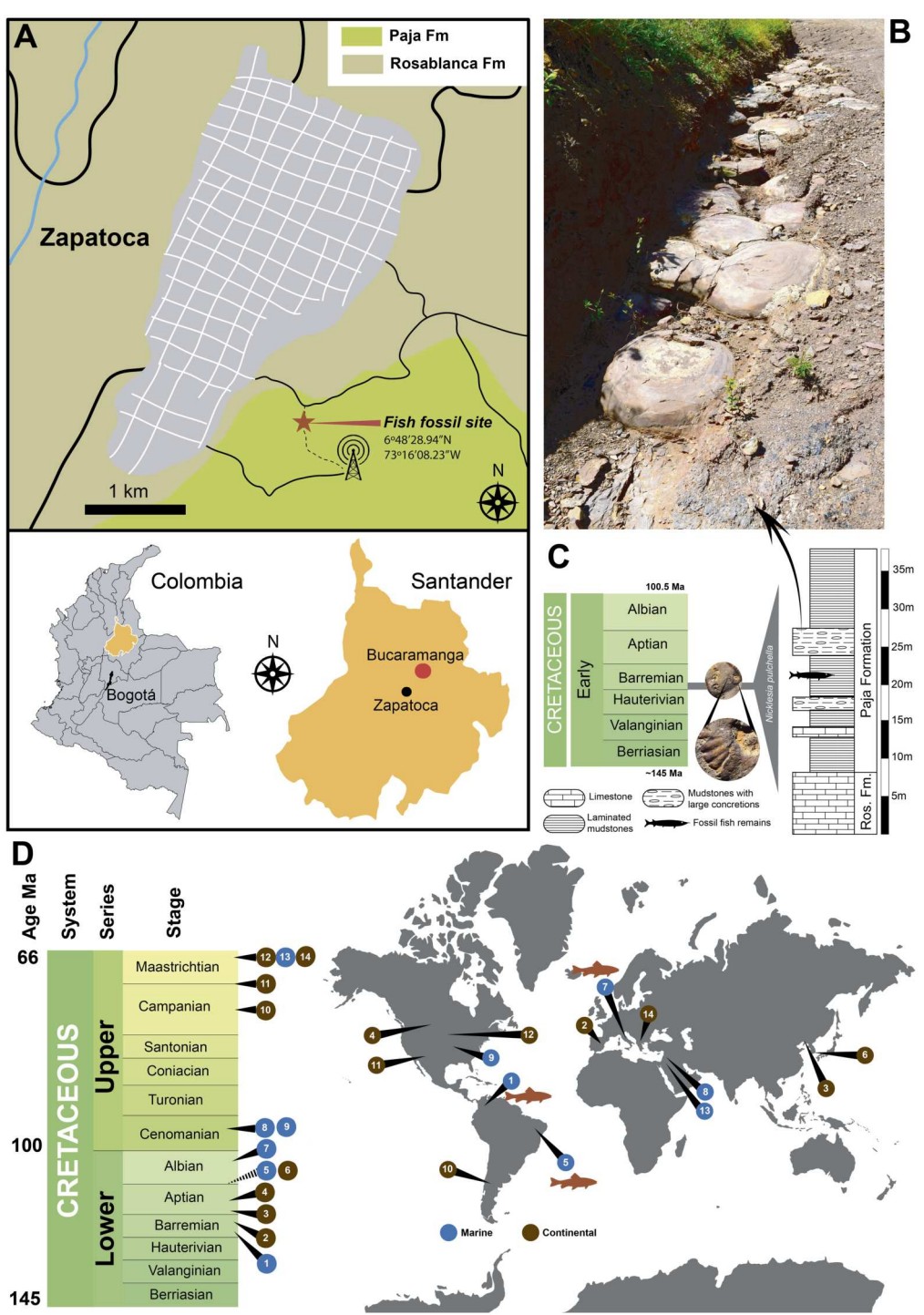

**Figure 1 Locality and other reported exceptionally preserved skin fossils from the Cretaceous.** (A) map of Colombia showing in orange the Santander department, and the fish fossil site (Zapalonga locality) very near Zapatoca. (B) outcrop view at the fish fossil site, showing the presence of mudstones and large concretions. (C) stratigraphic column along with Zapalonga locality, indicating the 

stereomicroscopes coupled with cameras. Measurements of the specimen were obtained using calipers, always wearing nitrile gloves during its manipulation. The specimen was scanned using computer tomography (CT-scan), Toshiba Aquilion at the Radiology unit of Hospital Méderi, Bogotá, with the following parameters: voltage 120 kV, exposure 225 mAs, and voxel size 350 $\mu$m.

**Transmitted and polarized light microscopy**. In order to observe and obtain microscopic details of the preserved 'skin', small pieces of approximately five mm$^3$ each were sampled and treated separated with HCl 25% for 24 h and EDTA 0.5 M pH 8.0 for 4 days changing daily to dissolve carbonate matrix and full demineralization. The isolated remains of 'skin' were rinsed 3 times with E-Pure water to remove HCl and EDTA, then were mounted in glass slides, observed and photographed using a Nikon ECLIPSE-80i transmitted-light microscope and an Olympus CX-31 polarized microscope. Samples were finally transferred to sterilized containers for Fourier-transform infrared spectroscopy (FTIR) analyses.

**FTIR spectroscopy**. Samples from an extant *Orechromis* sp. (Mojarra fish), and four samples from the UR-CP-0001 fossil fish ('skin' from HCl, EDTA treatments, 'skin' untreated and infilling matrix) were analyzed. The FTIR spectra were collected in the mid-infrared range of 4,000–600 cm$^{-1}$ wavelength using a Bruker Optics - ALPHA ZnSe FTIR spectrometer at the Biomedical Engineering Lab of Universidad de los Andes, Bogotá, Colombia. Between each analysis, the crystal and sample holder of the spectrometer were cleaned up with isopropanol and standardized with an "air" measurement in order to reduce rovibration absorptions of carbon dioxide present in the ambient air. Measurements were repeated twice for each of the samples. For the 'skin' untreated spectrum a deconvolution was performed for the 1,450–1,800 cm$^{-1}$ range in order to find out the specific peaks associated to the vibrational band frequencies of Amide I and II, similar as described in *Kong & Yu (2007)*.

**Scanning electron microscopy and elemental analysis (SEM-EDS)**. Four different regions of the fossil fish were sampled for Scanning Electron Microscope (SEM)-coupled with Energy Dispersive X-ray Spectroscopy (EDS) observation and characterization, taking ~5 mm$^3$ of each (scale 'skin', and two different regions of the infilling matrix exhibiting different coloration). Samples were mounted in sterile carbon stubs and storage in sterile boxes prior to the SEM-EDS analyses, which were performed at the Microscopy Core Facility of Universidad de los Andes, Bogotá, Colombia. Samples were analyzed without

adding any coating. Imaging and map elemental composition were obtained at 10 kV using a JEOL-JSM-6490 LV SEM, while the point elemental composition was performed at 10 kV using a TESCAN-Lyra3 SEM.

**Time of Flight Secondary Ions Mass Spectrometry (ToF-SIMS).** Two samples from the UR-CP-0001, an untreated (fresh) and an HCl treated were mounted in sterilized glass and sent to the Analytical Instrumentation Facility (AIF) of North Carolina State University, Raleigh, North Carolina. ToF-SIMS analyses were conducted using a TOF SIMS V (ION TOF, Inc. Chestnut Ridge, NY) instrument equipped with a $Bi_n^{m+}$ ($n = 1$–$5$, $m = 1, 2$) liquid metal ion gun, $Cs^+$ sputtering gun and electron flood gun for charge compensation. Both the Bi and Cs ion columns are oriented at 45° with respect to the sample surface normal, with at least two different regions of the sample being analyzed. The instrument vacuum system consists of a load lock for rapid sample loading and an analysis chamber, separated by the gate valve. The analysis chamber pressure is maintained below $5.0 \times 10^{-9}$ mbar to avoid contamination of the surfaces to be analyzed.

For high mass resolution spectra acquired in this study, a pulsed $Bi_3^+$ primary ion beam at 25 keV impact energy with less than 1 ns pulse width was used. An electron gun was used to prevent charge buildup on the insulting sample surfaces. The total accumulated primary ion dose for data acquisition was less than $1 \times 10^{13}$ ions/cm$^2$, an amount of ions which is within the static SIMS regime. The mass resolution on Si wafer is about ~8,000 m/$\Delta$m at 29AMU. For high lateral resolution mass spectral images acquired in this study, a Burst Alignment setting of 25 keV $Bi_3^+$ ion beam was used to raster a 500 $\mu$m by 500 $\mu$m area. The negative secondary ion mass spectra obtained were calibrated using $C^-$, $O^-$, $OH^-$, $C_n^-$, respectively. The positive secondary ion mass spectra were calibrated using $H^+$, $C^+$, $C_2H_3^+$, $C_3H_5^+$, $C_4H_7^+$.

## RESULTS

### Systematic Paleontology

Order ASPIDORHYNCHIFORMES Bleeker, 1859
Family ASPIDORHYNCHIDAE Nicholson and Lydekker, 1889
Genus and Species Indet. (Fig. 2)

Referred material.—UR-CP-0001, caudal portion of a fish, missing the fins.
Locality and Age.—Radio Lenguerke station antenna region, Zapalonga locality (6°48′28.94″N, 73°16′08.23″W, 1,703 m), southeast of Zapatoca, Santander Department, Colombia. The occurrence of the ammonoid *Nicklesia pulchella* (Fig. 1C) found in the same layer and concretions cropping out at this locality, indicates an early Barremian age for this locality following (*Patarroyo, 2009*; *Patarroyo, 2020*).
Remarks.—UR-CP-0001 is attributed to the Aspidorhynchidae family by the presence of rectangular high hypertrophied flank and nearly subquadrate scales covering the lateral and ventral sides of the trunk (*Brito, 1997*; *Cantalice, Alvarado-Ortega & Brito, 2018*) (Figs. 2G–2J). Although further taxonomic resolution is not possible owing to its

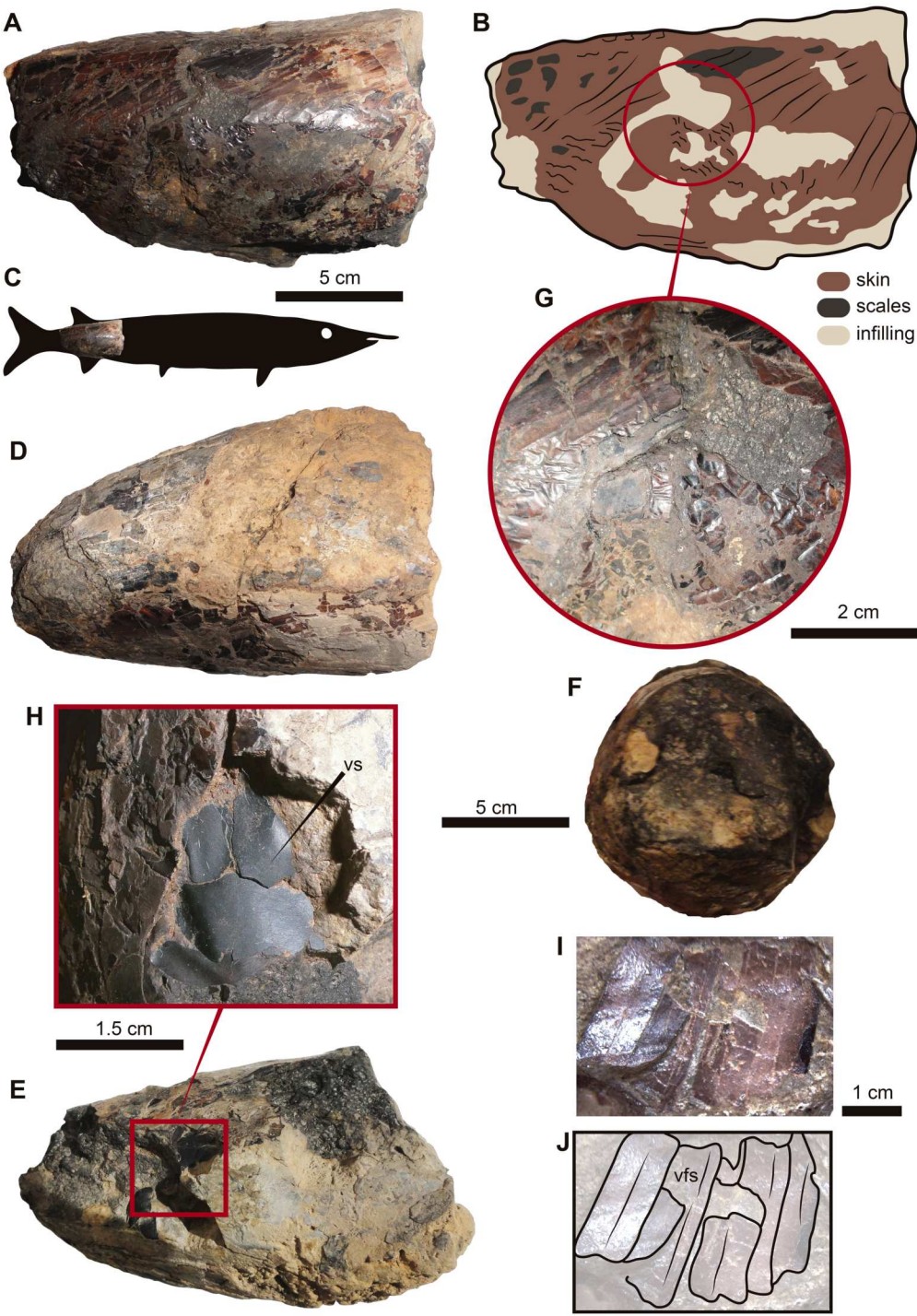

**Figure 2 UR-CP-0001, aspidorhynchid fossil fish specimen.** (A–B) right lateral view. (C) interpreted position of UR-CP-0001 in the body of an aspidorhynchid fish. (D) left lateral view. (E) ventral view. (F) posterior view, showing the naturally preserved original 3-D volume. (G) detail of the originally preserved 'skin' with wrinkles and marks. (H) View of some of the ventral scales (vs) preserved. (I–J) elongated ventral flank scales (vfs). five cm scale applies for A, D, E and F; two cm for G; 1.5 cm for H and one cm for I and J.

fragmentary preservation, the smooth surface of the flank scales resemble those of *Vinctifer comptoni* (see *Cantalice, Alvarado-Ortega & Brito, 2018*), suggesting the possibility that this organism represents a member of this taxon. Aspidorhynchids constitute an extinct basal teleostean group from the Middle Jurassic to Late Cretaceous fishes that were highly specialized and lived in shallow epicontinental marine environments throughout America, Europe, Australia, Africa, Antarctica, and Middle East (*Cantalice, Alvarado-Ortega & Brito, 2018*). The occurrence of the aspidorhynchid *Vinctifer* has been previously reported from exposures of the Paja Formation cropping out near Villa de Leyva, in the Department of Boyacá (*Noé & Gómez-Pérez, 2020*; *Schultze & Stöhr, 1996*).

**Description.** UR-CP-0001 represents a caudal portion of a fish preserved three-dimensionally (Figs. 2A–2D). The specimen is shaped like a truncated cone, which fits with the shape of caudal portions of other aspidorhynchids previously reported (Fig. 2C). Also the orientation of the scales impressions left on the skin exhibits a pattern typical of the caudal region (Fig. 2B).

The fossil has a length of 128.5 mm, an anterior height of 84 mm, and a posterior height of 40 mm. On the ventral surface there is a region that shows a scar that resembles the potential insertion of the anal fin. The edges of the specimen are completely eroded and no sign of bones is visible, which suggests that most of the anterior part of the specimen was probably lost prior the fossilization

Most of the lateral surfaces of the specimen bear a brown, wrinkled layer preserving 'skin' and covered in some places by rectangular black scales (Fig. 2B). These are particularly visible on the right side (Fig. 2G), whereas on the ventral side there are small, square marks similar to the ventral scales (Figs. 2H, 2I). There are no vertebrae or spines visible on the naturally broken anterior or posterior surfaces (Figs. 2E, 2F) nor are any visible internally in Computed Tomography (CT) of the specimen, which is infilled by a heterogeneous black-gray and yellow carbonate matrix (hereinafter infilling matrix) that is high-porosity in some regions and reacts to HCl (Video S1).

After demineralization with either HCl or EDTA (Figs. 3A, 3B) isolated pieces of 'skin' from fragments of fossil material (handled following aseptic techniques (see methods) and no glues or preservatives were applied) were observed under transmitted light microscope, and were shown to be formed by two distinct layers. Similar layers were observed in the dry skin of the extant *Orechromis* sp. (Mojarra fish) (Fig. 3C) together to some parallel lines similar to fibers observed in the extant and the fossil (Figs. 3D, 3F, 3G). The most basal layer is a thin semitransparent film-like sheet; this layer is covered by a brown to black organic patchy layer, in some degraded regions form irregular reticular pattern (Figs. 3I–3K). The basal semitransparent layer is quite flexible when wet, but becomes rigid and fragile when dried (Video S2). Under polarized light, the basal layer of the HCl-treated samples exhibits small granules having a first order of birefringence, indicating a potential phosphatic composition. The external organic brown layer covering this basal layer remains of the same color when the polarizer is rotated (Figs. 3L–3M). Pieces treated with EDTA showed higher degradation characterized by less and smaller fragments of both layers in contrast to those treated with HCl (Fig. S1). We consider that the external organic brown layer is consistent with the most exterior morphological feature of the skin, which is the

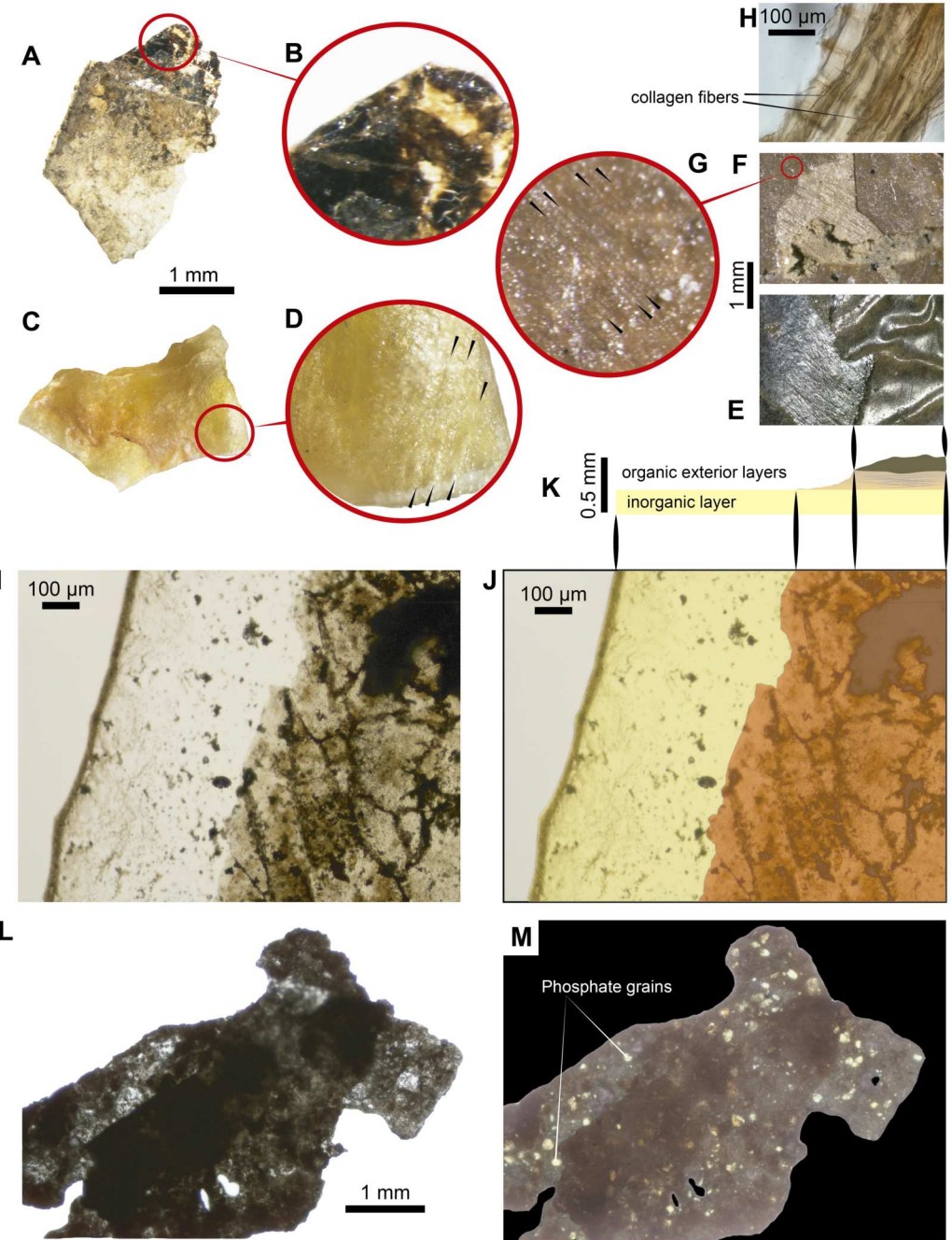

**Figure 3  Some 'skin' fragments after HCl treatment.** (A) Light micrograph of preserved 'skin' after treated with 15% HCl, without any infilling matrix left. (B) Enlargement of the organic patchy layer. (C–D) Fragment of the dry skin of the extant *Orechromis* sp. (Mojarra fish) exhibiting two layers, wrinkles and collagen fibers indicated by black arrows in d. (E) Wrinkled 'skin' of UR-CP-0001. (F–G) An UR-CP-0001 close-up of the two organic exterior layers and collagen fibers indicated by black arrows in g. (H) isolated tissue fragment after EDTA treatment under transmitted-light microscopy showing collagen fibers. (I–K) An UR-CP-0001 'skin' fragment under transmitted-light microscope, exhibiting the two distinct inorganic (base) and organic (exterior) layers. (L–M) An UR-CP-0001 'skin' fragment under transmitted-light (L) and polarized-light (M), showing low birefringence of the granular basal layer. One mm horizontal scale applies for A, C, L and M; one mm vertical scale for E and F.

epidermis (*Elliott, 2011*); also soft-tissue that are morphologically consistent with portions of the dermis were recovered after EDTA treatment, exhibiting collagen fibers (Fig. 3H).

**SEM-EDS results**. The untreated, uncoated skin is very smooth and uniform under SEM, which contrasts with the highly granular topography of the surrounding infilling matrix (Figs. 4A–4D). Point elemental analyses show predominant occurrence of carbon and nitrogen, with minor representation of calcium and phosphorus in the 'skin' layer (Fig. 4C). The infilling matrix contains predominantly calcium and carbon; no nitrogen was observed (Fig. 4D). Similar results were obtained using elemental mapping of the 'skin' and matrix (Figs. 4G–4L); however, nitrogen was not clearly observed.

**FTIR results**. The FTIR spectrum of the untreated 'skin' sample showed distinct peaks at 2,931 cm$^{-1}$, 1,740 cm$^{-1}$, 1,591 cm$^{-1}$ and around 1,120 cm$^{-1}$. The EDTA-treated sample showed high infrared absorption peaks at 1,703, 1,540 and 3,744 cm$^{-1}$ respectively (Fig. 5A). The HCl-treated sample showed absorption peaks at 1,724, 1,142, and 1,027 cm$^{-1}$ (Fig. 5A). The commercial extant fish skin sample (*Orechromis* sp. mojarra fish), exhibited two well defined regions of peaks at 1,746, 1,647, 1,559, and 1,117 cm$^{-1}$ and second one with peaks at 3,319 and 2,931 cm$^{-1}$. In contrast, the infilling matrix from UR-CP-0001 showed clear peaks at 1,428 cm$^{-1}$, 1,030 cm$^{-1}$ 876 cm$^{-1}$ and 711 cm$^{-1}$ (Fig. 5A).

**ToF-SIMS results**. ToF-SIMS analyses of both the untreated fossil 'skin' and the HCl-treated 'skin' show almost the same as each other negative and positive ions spectra (Fig. 6, Fig. S2); in particular, in abundance of CN$^-$ (Fig. 6C) and CNO$^-$ (Fig. 6F) negative ions; CH$_4$N$^+$ (Fig. 6D), C$_4$H$_8$N$^+$ (Fig. 6E), C$_2$H$_6$N$^+$ (Fig. 6G), and C$_3$H$_6$N$^+$ (Fig. 6H) positive ions were detected. All ions potentially derived from proteins are presented in Table 1, as well as all raw data obtained from ToF-SIMS analyses can be found in Data S1.

**Integrated compositional characterization of the 'skin' and comparisons.** As we showed using transmitted light, polarized light, and SEM-EDS microscopy (Figs. 3 and 4), the preservation of the 'skin' in UR-CP-0001 resulted from an organic and inorganic interaction forming two well defined layers (Figs. 3J, 3K), each of them exhibiting distinct physical and chemical characteristics. The basal layer is translucent, granular to film-like in appearance. This layer is interpreted as inorganic in composition, potentially phosphates, based on its birefringence pattern (Fig. 3M), the abundance of phosphorus showed by the EDS analysis (Fig. 4G, K-Phosphorus) together with the high absorbance peaks at 1,177 and 998 cm$^{-1}$ observed in the FTIR spectra. These peaks are particularly intense in the UR-CP-0001 sample (Fig. 5A), and were reported in an FTIR analysis of *Vinctifer comptoni* from the Cretaceous of Brazil (*Sousa Filho et al., 2016*). Similar peaks at this region have been interpreted as four infrared absorption bands of phosphate (vPO$_4$$^{3-}$ 1,120 cm$^{-1}$, v$_{3a}$ PO$_4$$^{3-}$1112 cm$^{-1}$, v$_{3c}$ PO$_4$$^{3-}$ 1,007 cm$^{-1}$ and v$_1$ PO$_4$$^{3-}$ 966 cm$^{-1}$) (*Lee et al., 2017*). Occurrence of phosphates and carbonates could be inferred from both SEM-EDS and FTIR analyses (Figs. 4K, 4L ; 5A) in the infilling matrix similar to the typical calcium carbonate FTIR spectrum (*Bosch-Reig, Gimeno-Adelantado & Moya-Moreno, 2002*). The more external layer of the 'skin' in UR-CP-0001 is brown to black, and is consistent with organic material when analyzed under polarized light (Fig. 3M). Its organic composition is supported by the SEM-EDS point and map analyses, which showed particularly high levels of carbon and nitrogen (Figs. 4C, 4H). Another remarkable finding that supports

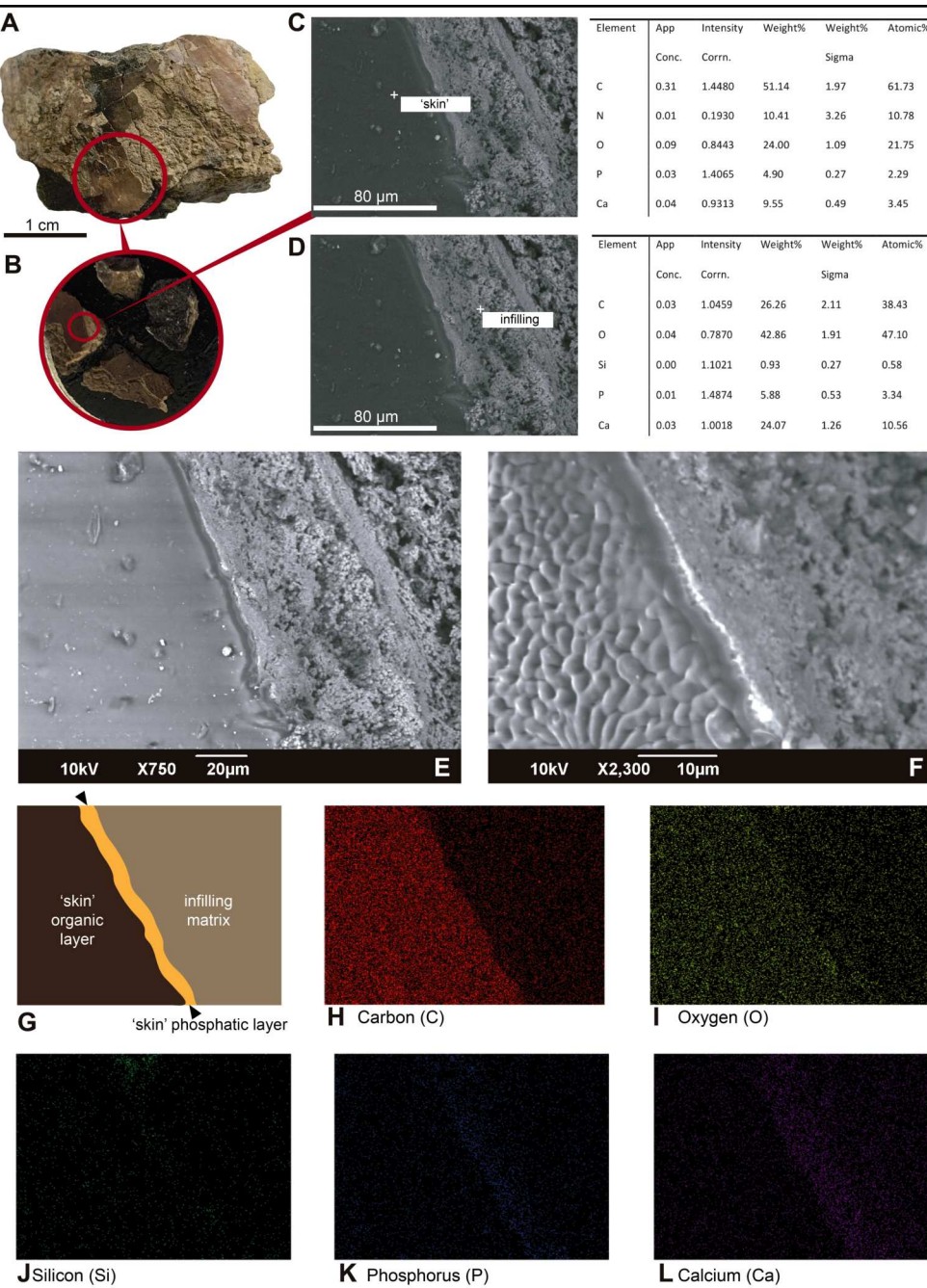

**Figure 4** SEM-EDS micrographs and elemental composition analyses of an untreated and uncoated fragment of UR-CP-0001 'skin'. (A) Sample from UR-CP-0001 that contains 'skin' and infilling matrix. (B) detail of samples mounted over the stub. (C) SEM micrograph with point EDS analysis in the 'skin' region, showing the abundant content of carbon and nitrogen, with less occurrence of calcium and phosphorous. (D) SEM micrograph with point EDS analysis in the infilling matrix, showing absence of nitrogene, dominance of carbon and calcium instead. (E) SEM micrograph of the 'skin'-infilling matrix contact before apply the EDS analysis. (F) Same micrograph as in (E) after EDS analysis, showing the extremely wrinkled organic surface of the 'skin', remaining intact the infilling matrix region. (G) Outline of the 'skin' organic and phosphatic layer, as well as the infilling matrix showed in e, which is the base of the elemental mapping. (H–L) Elemental mapping at 10 kV of the 'skin' infilling matrix region showing dominance of carbon (H) and oxygen (I) at the organic region, and phosphorus (K) at the boundary between the 'skin' and the infilling matrix; silicon (J) is very scarce in both regions, and of calcium (L) is highly abundant in the infilling-matrix.

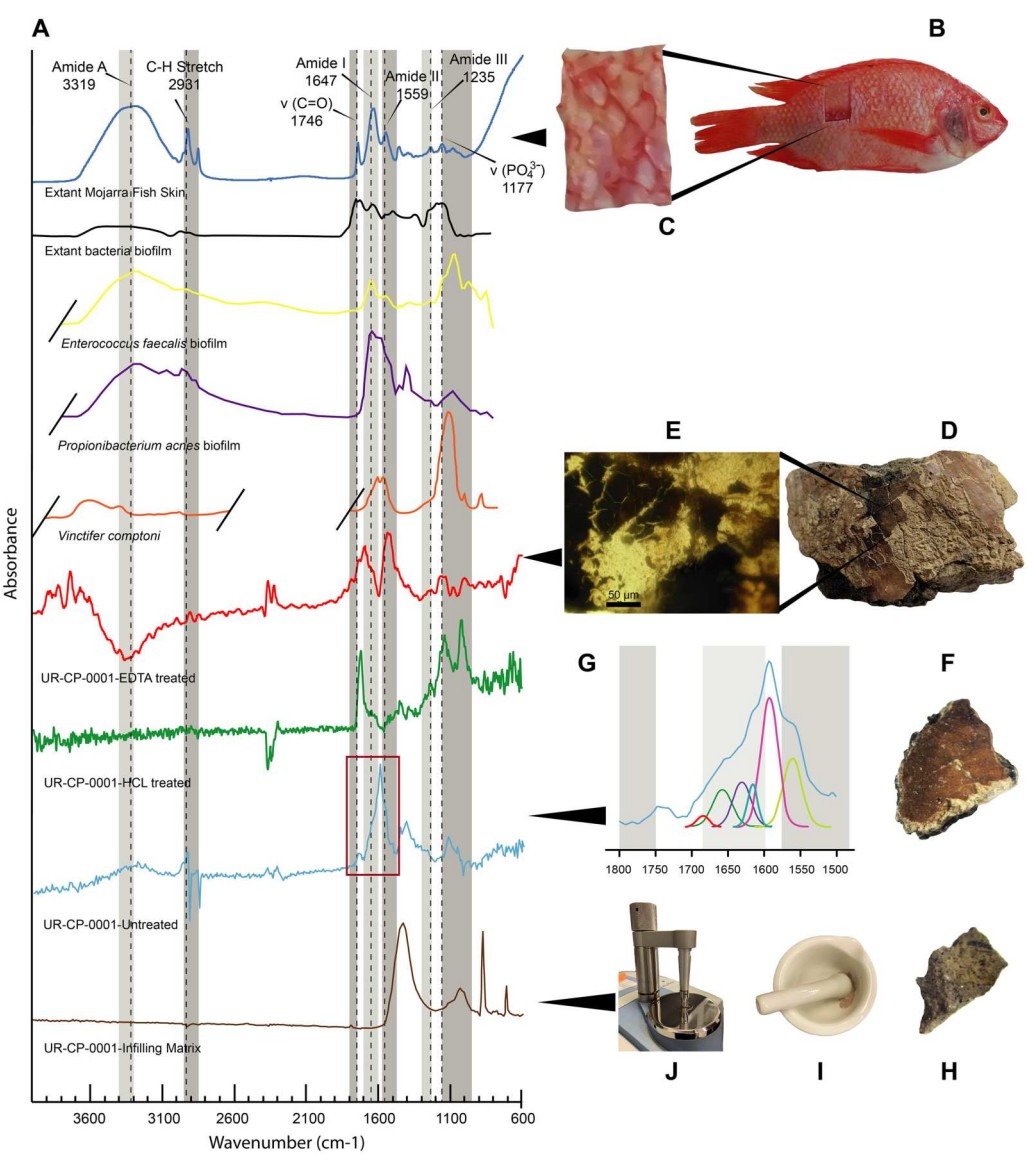

**Figure 5** **FTIR analyses of UR-CP-0001 and the extant *Orechromis* sp.** (A) Composite FTIR spectra (absorbance vertical axis, wavenumber horizontal axis) of different samples: *Orechromis* sp. (Mojarra fish) (dark blue line) with interpretation of typical proteinaceous compounds (Amide A, I, II, III, v(C=O), C-H stretch and a phosphate) with gray bands showing potential ranges based on *Boatman et al. (2019)*; *Kong & Yu (2007)*; and *Lee et al. (2017)*; an extant bacteria biofilms (black, yellow and purple lines) taken and redraw from *Lee et al. (2017)* and *Lindgren et al. (2011b)*; *Vinctifer comptoni* (orange line) from the Cretaceous of Brazil, taken and redrawn from *Sousa Filho et al. (2016)*; UR-CP-0001 aspidorhynchid fossil fish 'skin' treated with EDTA (red line); treated with HCl (green line); untreated (light blue line); and UR-CP-0001 infilling matrix (brown line). (B) Skin sample from *Orechromis* sp. (Mojarra fish) used for the FTIR analysis and close-up of the skin sample analyzed from this specimen (C). (D) The region from which the 'skin' sample of UR-CP-0001 was taken, and a close-up of the 'skin' fragment after EDTA treatment under a transmitted light microscope (E). (F) UR-CP-0001 sample used for the untreated analysis and a close-up of the FTIR vibrational bands (red rectangle) in (A) after deconvolution (G). (H) UR-CP-0001 infilling matrix sample and how it was grinded using a sterilized mortar and pestle (I) and placed in the FTIR machine (J).

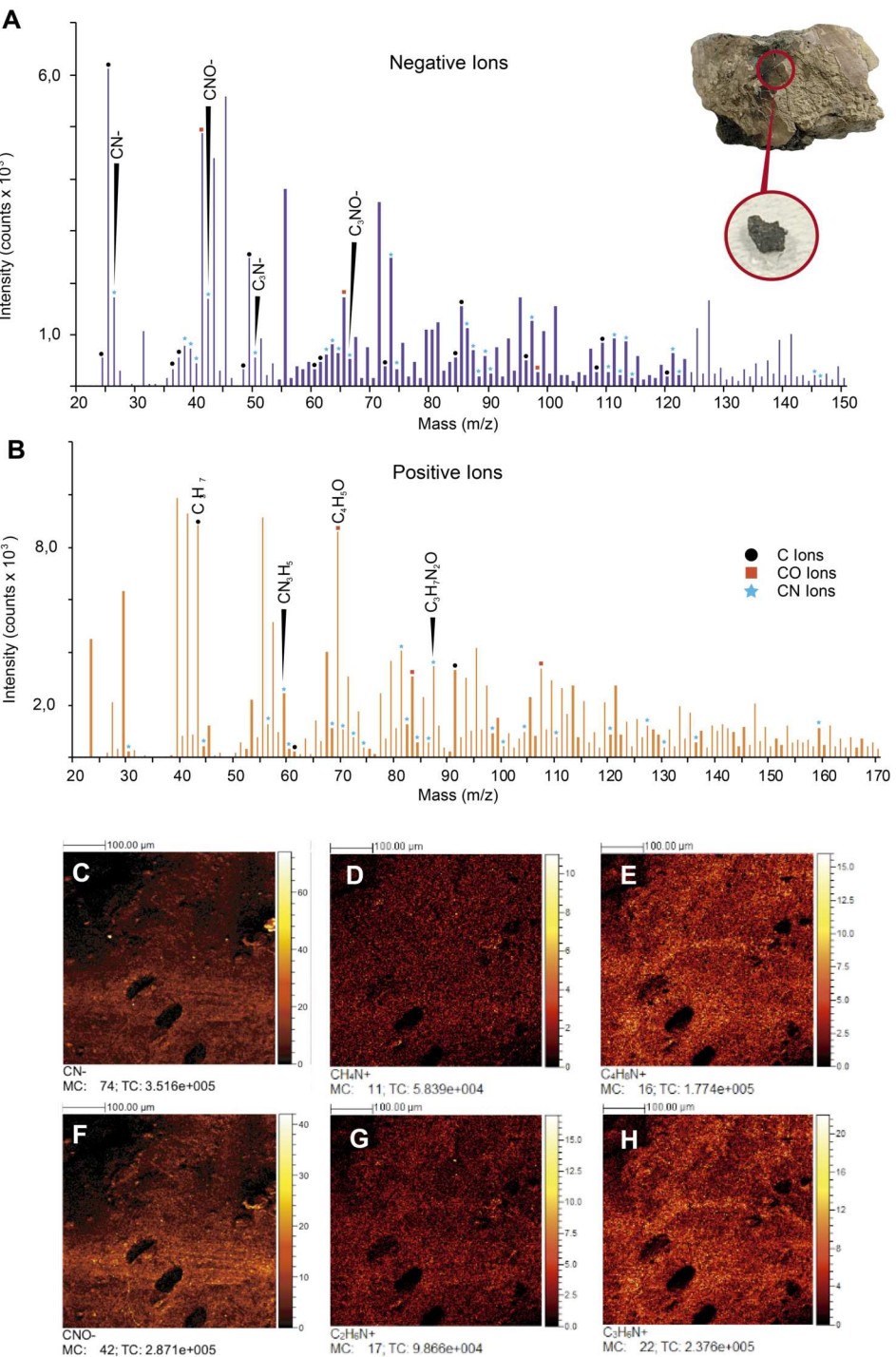

**Figure 6 ToF-SIMS analyses of UR-CP-0001 'skin'.** (A–B) Negative and Positive ion ToF-SIMS spectrum of UR-CP-0001 untreated sample (see circular photo of the sample), typical organic compounds occur in high intensities in both ions (raw data presented in Data S1). (C-H) ToF-SIMS images showing the distribution of ions CN–(C), CH4N+ (D), C4H8N+ (E), CNO–(F), C2H6N+ (G) and C3H6N+ (H).

**Table 1** Species tentative assignments and m/z values for peaks in both positive and negative ToF-SIMS spectra from UR-CP-0001 and its possible organic source based on *Samuel et al. (2001)*, *Brüning et al. (2006)*, *Lindgren et al. (2018)* and *Lindgren et al. (2012)*.

| Tentative assignment | Theoretical mass | Fossil sample (UR-CP-0001) | Associated organic compound |
|---|---|---|---|
| CN- | 26.00 | 25.997 | Melanin |
| $CH_4N$ | 30.036 | 29.998 | Glycine |
| CNO- | 42.00 | 42.001 | Melanin |
| $C_3H_7$ | 43.03 | 42.998 | Leucine |
| $C_2H_6N$ | 44.053 | 43.999 | Alanine |
| $C_3N$- | 50.00 | 50.000 | Melanin |
| $C_3H_6N$ | 56.05 | 55.996 | Lysine |
| $CN_3H_5$ | 59.05 | 59.001 | Arginine |
| $C_2H_6NO$ | 60.045 | 59.999 | Serine |
| $C_2H_5S$ | 61.01 | 60.998 | Methionine |
| $C_3NO$- | 66.00 | 65.996 | Melanin |
| $C_4H_6N$ | 68.05 | 67.996 | Proline |
| $C_4H_5O$ | 69.03 | 68.998 | Threonine |
| $C_4H_8N/C_3H_4NO$ | 70.068 | 70.000 | Proline |
| $C_4H_{10}N$ | 72.084 | 71.997 | Valine |
| $C_3H_8NO$ | 74.063 | 73.998 | Threonine |
| $C_5N$-$/C_2H_2O_3$ | 74.00 | 74.000 | Melanin |
| $C_4H_5N_2$ | 81.04 | 80.996 | Histidine |
| $C_4H_6N_2$ | 82.05 | 81.998 | Histidine |
| $C_5H_7O$ | 83.09 | 83.000 | Valine |
| $C_5H_{10}N$ | 84.085 | 84.003 | Lysine |
| $C_5H_{12}N/C_4H_8NO$ | 86,064/86,101 | 86.002 | Hydroxyproline/Leucine |
| $C_3H_7N_2O$ | 87.05 | 86.999 | Aspargine |
| $C_7H_7$ | 91.05 | 90.998 | Phenylalanine |
| $C_4H_4NO_2$ | 98.02 | 97.999 | Aspargine |
| $C_4H_{10}N_3$ | 100.088 | 99.999 | Arginine |
| $C_4H_{10}NS$ | 104.05 | 104.002 | Methionine |
| $C_7H_7O$ | 107.048 | 106.999 | Tyrosine |
| $C_8N/C_9H_2$ | 110.075 | 109.999 | Histidine |
| $C_8H_{10}N$ | 120.084 | 120.000 | Phenylalanine |
| $C_5H_{11}N_4$ | 127.1 | 126.997 | Arginine |
| $C_9H_8N$ | 130.068 | 130.003 | Tryptophan |
| $C_8H_{10}NO$ | 136.082 | 136.005 | Tyrosine |
| $C_{10}H_{11}N_2$ | 159.04 | 159.00 | Tryptophan |

the organic composition of this layer is its morphological change after being exposed to 10 kV for mapping EDS analysis becoming highly corrugated (Figs. 4E, 4F), which typically happens to uncoated organic tissue or structures under high voltage in SEM similar as degradation of non-conductive materials (*Kersten, 2009*).

FTIR analysis confirmed that the carbon rich layer we found with the EDS is composed of, organic residues, particularly the C-H stretch and $v(C=O)$ peaks around 2,931 and

1,737 respectively (Fig. 5A), which are commonly found in collagen I (*Belbachir et al., 2009*; *Jeevithan et al., 2014*; *Valenzuela-Rojo, López-Cervantes & Sánchez-Machado, 2018*) and keratin (*Chandini et al., 2017*; *Estévez-Martínez et al., 2013*); highly abundant proteins found in the scales and skin of fishes (*Bhagwat & Dandge, 2016*; *Elliott, 2011*). Amide A, I, II, and III, C-H stretch and v(C=O) peaks were clearly observed in the FTIR of the extant *Orechromis* sp. (mojarra fish) skin used as standard for comparison (Fig. 5A). Peaks potentially corresponding to Amide I and II were also found in the deconvoluted spectrum of the 'skin' untreated sample (Fig. 5D), falling inside the range of vibrational bands as product of possible diagenetic alterations of the original organic compounds, similar as occurs in FTRI analyses of modern proteins (*Kong & Yu, 2007*). We exclude a potential bacterial origin of the organic component of the 'skin' in UR-CP-0001 because FTIR spectra lack of the characteristic broad infrared absorption band of hydroxyl group (-OH) of polysaccharides at 3,700–3,100 $cm^{-1}$ (*Lee et al., 2017*; *Lindgren et al., 2011b*). ToF-SIMS results of the two samples of UR-CP-0001 analyzed also show the occurrence of molecular organic fragments, including the positive $CH_4N^+$ (Fig. 6D), $C_4H_8N^+$ (Fig. 6E), $C_2H_6N^+$ (Fig. 6G), $C_3H_6N^+$ (Fig. 6H) and $C_7H_7O^+$ which are typical residues of glycine, alanine, proline and tyrosine constituents of collagen and fibronectin (*Brüning et al., 2006*; *Henss et al., 2013*). Two other ions that support potential organic preservation in the 'skin' of UR-CP-0001 are $CN^-$ (Fig. 6C) and $CNO^-$ (Fig. 6F) negative ions particularly abundant in melanosomes and melanin (*Lindgren et al., 2018*; *Lindgren et al., 2012*), and although we can not reject at this point that they could be from another source, our hypothesis seems to be plausible. A complete tentative assignment of ions derived from proteins based on m/z values in UR-CP-0001 samples and theoretical mass is presented in Table 1. We exclude a potential mineralized biofilm source of protein residues based on the FITR spectra (Fig. 5A) and the absence of any morphological features associated to bacteria origin (filaments or spheres) (*Kaye, Gaugler & Sawlowicz, 2008*; *Schweitzer, Moyer & Zheng, 2016*).

The preservation of the 'skin' in UR-CP-001 is also supported by its morphological corrugated macroscopic appearance (Figs. 2G, 3E) resembling a phenomenon that occurs to the skin from extant fishes where an absence of scales leaves the skin without an external support structure, make it more susceptible to wrinkling under a compression stress (*Vernerey & Barthelat, 2014*), due to dehydration or in a post mortem deformation (*Lindgren et al., 2018*) (Figs. 3C, 3E). Additionally, collagen fibers were observed in both UR-CP-0001 'skin' and the dehydrated skin from extant *Orechromis* sp. (mojarra fish) also to microscopic level after EDTA demineralization of 'skin' (Figs. 3D, 3G, 3H) supporting the interpretation of UR-CP-0001 as an exceptional preserved fossilized skin.

## DISCUSSION

Aspidorhynchid fishes had widespread geographic and temporal distribution with fossils reported in all continents from the Middle Jurassic to Late Cretaceous (*Brito, 1997*). Specimen UR-CP-0001 represents the earliest known record for an aspidorynchid in Colombia, extending the temporal range from the Aptian (*Schultze & Stöhr, 1996*) to Barremian. Once again, a peri-Gondwanan distribution of *Vinctifer* (Fig. S3) is confirmed here, as UR-CP-0001 potentially belongs to this genus (see Remarks).

Vibrational spectroscopic techniques such FTIR demonstrates its reliability to understand fossil preservation mechanisms, due to its sensitiveness to organic functional groups and phosphates thought high peak bands (*Lafuente-Diaz et al., 2020*; *Olcott Marshall & Marshall, 2015*). However, due to noise signals, a deconvolution was needed to unveil masked absorbance peaks from the raw data. ToF-SIMS also give more resolution to identify the nature of preserved components. These kind of analysis has demonstrate to be trustful for inferences about preservation mechanisms and track the origin of the preserved molecules (*Bezerra et al., 2020*; *Lafuente-Diaz et al., 2020*).

Although it is hard to reconstruct the complete chain of taphonomical events that occurred to UR-CP-0001, we hypothesize that besides fragmentation and fins disarticulation without losing the conical shape of its caudal region, the nature of its scales and skin played a key role in its preservation. The presence of scales and the thickness of the fossilized 'skin' suggest a possible mechanism of preservation that we call a "microsandwich effect", which could apply to many other fragmentary fossil fishes that have not been studied for molecular paleontology. Scales may have acted as an external barrier against bacteria and other environmental decay accelerators, which could decompose the integument. Simultaneously, the basal layer became enriched in phosphate, possibly resulting from the degradation of phosphate containing organic compounds from the dermis itself, as has been reported in other fossilized skin from vertebrates (*McNamara et al., 2009*), at the same time this layer may have acted as an internal barrier, creating an encapsulating environment for the integument. These local biogeochemical interactions would favor not only preservation of the general morphology of the skin, but also some of their soft-tissue structures and residues of the original biomolecules by geopolymerization (*Lindgren et al., 2018*). Another factor that potentially played a key role in the preservation of the 'skin' in UR-CP-0001 was the burial environment conditions, dominated by organic-rich shale interval showing characteristics of oxygen depleted conditions at the lower segment of Paja Formation in this region (*Gaona-Narvaez, Florentin & Etayo-Serna, 2013*). Microcrystalline minerals like clays and shales have extremely large surface area to volume ratios, and are usually charged, both of which favor adsorption and inactivation of degrading enzymes, similar been proposed for the exceptional preservation of Burgess Shale fossils (*Butterfield, 1990*).

Our results imply that the Paja Formation could be potentially considered as the third locality in South America where exceptional preservation in fishes have been reported, alongside of the Brazilian Romualdo and Crato Formations, where the preservation mechanisms is well known (*Osés et al., 2017*). The mechanism of preservation proposed here, as well as other recent work (*Lindgren et al., 2018*) increases the number of potential scenarios for preservation of cellular-to-subcellular soft tissue morphology in fossils additional to oxidative depositional environments (*Wiemann et al., 2018*), where iron play a key role (*Schweitzer et al., 2014*). As we showed in here, iron was not detected in UR-CP-0001, suggesting that in molecular paleontology studies there will be always exceptions to those formulated general trends and factors favoring preservation in deep time, and that each case and fossil site needs to be considered with its own particularities.

## CONCLUSIONS

Exceptional preserved 'skin' from an aspidorhynchid fish represents the first report of soft tissue preservation in vertebrates from the Early Cretaceous in north South America. Morphological comparisons and molecular analyses present several similar features between the extant fish skin and the fossilized specimen. Molecular analyses also provide evidence of possible proteinaceous residues preserved in the fossilized skin, which is supported by vibrational peaks associated with Amide I and II in the FTIR spectra and signals that can be associated to aminoacids like Glycine and Lysine. Because of the limitation in the project funding, future analyses should be focused on immunohistochemistry, testing specific fish skin antibodies and other mass spectrometry techniques including LC-MS/MS to confirm the preservation of original proteinaceous components.

**Institutional abbreviation**

**UR-CP**      paleontological collection, Facultad de Ciencias Naturales y Matemáticas, Universidad del Rosario, Bogotá, Colombia.

## ACKNOWLEDGEMENTS

We thank to E Realpe for allowing us to use the stereomicroscope. Special thanks to A Link for helping us with the logistics. Thanks to M Negrete and the radiology team at the Hospital Méderi for access to the CT-scan. Thanks to M López and H Pinto from the Universidad de los Andes, Bogotá for the scanning this fossil with the SEM-EDS and the analyses performed with the FTIR spectrometer. Thanks to Y Rojas for allow us the use of the polarized transmitted light microscope at Universidad de los Andes. Thanks to A Forero and L Daza for assistance during lab preparation of the samples. Thanks to C Zhou for the ToF-SIMS analyses. Special thanks to M Schweitzer and JA Wilson, as well as to M. Benton and an anonymous reviewer for valuable comments on the manuscript.

### Funding

Funding for this project was granted to Edwin Alberto Cadena from Universidad del Rosario, Capital Semilla grants program 2019 and Fondos de Arranque 2018: (Code IV-TFA022). The funders had no role in study design, data collection and analysis, decision to publish, or preparation of the manuscript.

### Grant Disclosures

The following grant information was disclosed by the authors:
Edwin Alberto Cadena from Universidad del Rosario, Capital Semilla grants program 2019. Fondos de Arranque 2018: (Code IV-TFA022).

### Competing Interests

The authors declare there are no competing interests.

## Author Contributions

- Andrés Alfonso-Rojas conceived and designed the experiments, performed the experiments, analyzed the data, prepared figures and/or tables, authored or reviewed drafts of the paper, photos, and approved the final draft.
- Edwin-Alberto Cadena conceived and designed the experiments, performed the experiments, analyzed the data, prepared figures and/or tables, authored or reviewed drafts of the paper, fieldwork, and approved the final draft.

## Field Study Permissions

The following information was supplied relating to field study approvals (i.e., approving body and any reference numbers):

Field experiments were approved by the Comité de ética and the Dirección de Investigaciones of the Universidad del Rosario (IV-FCS018).

## Data Availability

Raw data is available in the Supplementary Files.

## Supplemental Information

Supplemental information for this article can be found online at http://dx.doi.org/10.7717/peerj.9479#supplemental-information.

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
