# Peer review of "Exceptionally preserved ‘skin’ in an Early Cretaceous fish from Colombia"

_PeerJ, doi:10.7717/peerj.9479_

## Round 0.1 · original submission · Major Revisions

I have two reviews of your manuscript. Both reviewers agree that you have interesting data here. However, several issues should be considered before going ahead. In the first place, I would like you to stress the theme of the paper. Our first reviewer gives you some possibilities to address this point.

Our second reviewer is concern about you not quoting several pertinent papers, using an outdated stratigraphy, misplaced localities, among other points. Please, take all these suggestions in full consideration since all of them are delicate points.

·

Basic reporting

This is all fine.

Experimental design

This all seems good.

Validity of the findings

The authors use standard analytical methods, which are clearly explained, and I think all conclusions are reasonable. In the Discussion, perhaps say more about the usefulness and trustworthiness of the various methods, especially FTIR (where identification of peaks is a bit of an art).

Additional comments

The first sentence isn’t helpful – why does it matter that it’s from South America? The first sentence should be of global interest – what’s unique about the specimen? Maybe the best that can be said is that ‘Exceptional preservation of soft tissues in fishes is well known from a number of localities (Santana, Liaoning…), and we present here the first such report from a new location… so establishing the Paja Formation fish beds as a Lagerstätte.

The initial part of the ‘Discussion’ (lines 230-288) is Results, so can you move it back to ‘Results’ and integrate with the existing text there.

Then the ‘Discussion’ will focus on the exceptional fossilization and the model for why soft tissues were preserved. This needs more comparison with recent papers on b Santana, Liaoning, and other such materials. Also explore the reliability of FTIR and the other methods used – are they entirely reliable?

70: couple = a few
103: clean = cleaned
225: aspidorinchid – aspidorhynchid
281: a phenomena = a phenomenon
319: Aspidorhynchid = aspidorhynchid

Reviewer 2 ·

Basic reporting

The ms. by Alfonso-Royas and Cadena describe a rare fossilization condition within vertebrates: the preservation of soft tissue. Overall, the ms. is well written and pretty straight forward: they have found a quite incomplete specimen but manage to find out that it shows well preserved portions with soft tissue and proceeded with detailed analysis that are not commonly done in fossils. This alone is warrant publication. There are some typos here and there (e.g., Brasil), but nothing of great concern.

The main point here is that the literature regarding soft tissue in fossil vertebrates of South America is not acknowledged.

Experimental design

Going through their study, there is nothing that I could identify or observe that would not fit to their interpretations.

Validity of the findings

As pointed out above, there are not many studies like the ones addressed in this paper.

Additional comments

The ms. by Alfonso-Royas and Cadena describe a rare fossilization condition within vertebrates: the preservation of soft tissue. Overall, the ms. is well written and pretty straight forward: they have found a quite incomplete specimen but manage to find out that it shows well preserved portions with soft tissue and proceeded with detailed analysis that are not commonly done in fossils. This alone is warrant publication. There are some typos here and there (e.g., Brasil), but nothing of great concern.
Going through their study, there is nothing that I could identify or observe that would not fit to their interpretations. There are some main points that the authors should correct despite being peripherical to they study.
1) The stratigraphy they are using from Brazilian deposits is outdated. The former Santana FM is now recognized as Santana Group and the deposit they point out with well preserved fish fossils (not only, see below) is now regarded as the Romualdo FM. There are several papers pointing this out - perhaps the authors wish to consult Kellner et al. 2013 (Anais da Academia Brasileira de Ciências, 85(1): 113-135; doi: 10.1590/S0001-37652013000100009) and/or the bibliography cited therein.
2) Please make sure that all other localities you pointed out in your figure 1 are in the right place. For example, the Brazilian deposit (Santana = Romualdo FM) is in the northeastern portion of the country not in the center.
3) Right in the abstract the authors mention that "preserved in fossil vertebrates from tropical South America are barely known". They than proceed with several inserting and relevant examples from different deposits in the Introduction. However, they should be aware of the several superb specimens from Brazil and China that have been in the literature, including soft tissue from fishes, dinosaurs and pterosaurs from the Romualdo Formation (former Romualdo member of the Santana Formation) as well as the material from several deposits of China, that include even "hair-like structures" such as pycnofibers. For all the material of the Romualdo FM with dermis, epidermis, muscle fibers and potential blood vessels (from dinosaurs such as Santanaraptor, pterodactyloid pterosaurs, as well as fishes) should be cited at least in the introduction since their material is from South America.

4) How sure are they that they have the caudal portion of a fish? This is not clear from the picture. The identification as and aspidorhynchid seems to be correct judging from the scales. They should comment if the edges of the specimen are completely preserved and that there is no indication that parts of the dorsal and ventral portion of the piece they got were not decomposed prior to fossilization or eroded away after fossilization.

There are some additional comments that they authors should address as they see fit:
4) l. 39 - Please acknowledge the specimens from China that include feathers, pycnofibers.

5) ls. 74-76 - It is nice the caution collecting the material. But they authors might like to disclose why this particular specimen was collected with that much care, a quite unusual procedure regarding fossil vertebrates. Did they know in advance the potential for finding such important material?

6) l. 148 - indet.

7) l. 150 - is this the first specimen of a new collection? This might be mentioned in the introduction - that the authors are building a new collection.

---

## Round 0.2 · Minor Revisions

Thank you for considering the reviewers'suggestions. A few more points require further attention. Please, spell out FTIR the first time you used it.
I think that the last sentence of your conclusions is too general to be considered as obtained for this paper, please delete it.

---

## Round 0.3 · accepted · Accept

Thank you for having my suggestions into account. We are ready to go ahead.